# Eosinophils of patients with localized and diffuse cutaneous leishmaniasis: Differential response to *Leishmania mexicana*, with insights into mechanisms of damage inflicted upon the parasites by eosinophils

Norma Salaiza-Suazo[1], Roxana Porcel-Aranibar[1], Isabel Cristina Cañeda-Guzmán[1], Adriana Ruiz-Remigio[1], Jaime Zamora-Chimal[1], José Delgado-Domínguez[1], Rocely Cervantes-Sarabia[1], Georgina Carrada-Figueroa[2], Baldomero Sánchez-Barragán[3], Victor Javier Leal-Ascencio[4], Armando Pérez-Torres[5], Héctor A. Rodríguez-Martínez[1], Ingeborg Becker[1]*

1 Facultad de Medicina, Unidad de Investigación en Medicina Experimental, Universidad Nacional Autónoma de México, Ciudad de México, México, 2 División Académica de Ciencias de la Salud, Universidad Juárez Autónoma de Tabasco (UJAT), Tabasco, México, 3 Escuela de Medicina, Universidad Juárez Autónoma de Tabasco (UJAT), Tabasco, México, 4 Hospital Regional de Alta Especialidad Dr. Juan Graham, Secretaría de Salud del Estado de Tabasco, Villahermosa, Tabasco, México, 5 Departamento de Biología Celular y Tisular, Facultad de Medicina, Universidad Nacional Autónoma de México, Ciudad de México, México

* becker@unam.mx

## Abstract

Eosinophils are mainly associated with parasitic infections and allergic manifestations. They produce many biologically active substances that contribute to the destruction of pathogens through the degranulation of microbicidal components and inflammatory tissue effects. In leishmaniasis, eosinophils have been found within inflammatory infiltrate with protective immunity against the parasite. We analyzed the responses of eosinophils from patients with localized (LCL) and diffuse (DCL) cutaneous leishmaniasis, as well as from healthy subjects, when exposed to *Leishmania mexicana*. All DCL patients exhibited blood eosinophilia, along with elevated eosinophil counts in non-ulcerated nodules. In contrast, only LCL patients with prolonged disease progression showed eosinophils in their blood and cutaneous ulcers. Eosinophils from DCL patients secreted significantly higher levels of IL-6, IL-8, and IL-13, compared to eosinophils from LCL patients. Additionally, DCL patients displayed higher serum levels of anti-*Leishmania* IgG antibodies. We also demonstrated that eosinophils from both LCL and DCL patients responded to *L. mexicana* promastigotes with a robust oxidative burst, which was equally intense in both patient groups and significantly higher than in healthy subjects. Coincubation of eosinophils (from donors with eosinophilia) with *L. mexicana* promastigotes *in vitro* revealed various mechanisms of parasite damage associated with different patterns of granule exocytosis: 1) localized degranulation on the parasite surface, 2) the release of cytoplasmic membrane-bound "degranulation sacs" containing granules, 3) release of eosinophil extracellular traps containing DNA and granules with major basic protein. In conclusion, eosinophils damage *L. mexicana* parasites through

**Data Availability Statement:** All relevant data are within the manuscript and its Supporting Information files.

**Funding:** This work was supported by UNAM-PAPIIT IG201221 and CONACyT-Fronteras 6682. The funders had no role in study design, data collection and analysis, decision to publish or preparation of the manuscript.

**Competing interests:** The authors have declared that no competing interests exist.

the release of granules via diverse mechanisms. However, despite DCL patients having abundant eosinophils in their blood and tissues, their apparent inability to provide protection may be linked to the release of cytokines and chemokines that promote a Th2 immune response and disease progression in these patients.

## Introduction

In humans, *Leishmania mexicana* can cause localized cutaneous leishmaniasis (LCL) and diffuse cutaneous leishmaniasis (DCL). The most common clinical form of the disease occurring in over 90% of the cases is LCL. It is characterized by ulcers at the sites of parasite inoculation, accompanied by a robust cellular immune response and a scarce number of parasites within the lesions. Patients with LCL generally experience a relatively benign clinical course [1]. Less than 1% progress to DCL, which is characterized by multiple non-ulcerated nodules containing heavily parasitized macrophages that cover large areas of the body. These patients exhibit an ineffective cell-mediated immune response against *Leishmania* parasites, leading to uncontrolled and potentially fatal progression [2]. In Mexico, 500–1000 new cases of cutaneous leishmaniasis are diagnosed every year [3].

In the early stages of infection, the inflammatory infiltrate includes neutrophils, macrophages, and eosinophils. Under normal conditions, eosinophils constitute only 1–5% of circulating leukocytes, although they can also be found in tissues as resident cells. Eosinophils are well equipped to combat pathogens, such as helminths, and their numbers increase during infection [4]. Eosinophils play key roles in immune modulation, tissue remodeling, and repair, primarily through the secretion and degranulation of inflammatory mediators, including cytokines and chemokines. However, their effector mechanisms can also inadvertently damage bystander cells and tissues [4,5].

During the acute phase of leishmaniasis, eosinophils can be present in skin lesions, often alongside mast cells. Furthermore, a close association between eosinophils and parasitized macrophages has been reported in chronic lesions, suggesting that eosinophils may contribute to parasite destruction through cooperation with macrophages [6]. Tissue eosinophilia has been observed in DCL patients, with evidence of parasitized and lysed eosinophils and dispersion of their granules near parasitized macrophages [7]. Additionally, both in the mouse model of leishmaniasis and in humans with chronic phases of the disease, eosinophils are frequently found in granulomas [8–10]. One of the mechanisms through which eosinophils assist in controlling the parasite is the production of hydrogen peroxide, which has a detrimental effect on *Leishmania amazonensis* [11]. However, the precise physiological role of eosinophils in leishmaniasis remains incompletely understood.

In this study, we conducted a comparative analysis of the *in vitro* response of eosinophils isolated from LCL and DCL patients, following their co-incubation with *L. mexicana*. Our analysis included the measurement of IL-6, IL8 and IL-13 production, as well as assessment of their oxidative burst.

Additionally, we co-incubated eosinophils from non-*Leishmania* eosinophilic subjects with *L. mexicana* promastigotes and analyzed their defense mechanisms against the parasites. Our findings reveal that eosinophils employ various defense mechanisms when confronted with *L. mexicana*. These mechanisms include phagocytosis, contact degranulation, the release of granules directly onto the parasite, expulsion of "degranulation sacs" containing granules surrounded by a membrane, which are released into the immediate vicinity of the parasites, and

the release of eosinophil extracellular traps (EETs) containing DNA and granules rich in major basic protein (MBP). All of these mechanisms result in irreversible damage to the parasite. Our data provide new insights into the innate defense mechanisms exerted by eosinophils against *L. mexicana*, as well as differences in the eosinophil response between LCL and DCL patients.

## Materials and methods

### Ethics statement

The study received approval from the Ethics and Research Committees of the Facultad de Medicina, UNAM (Universidad Nacional Autónoma de México) with reference FM/DI/088/2017. We adhered strictly to the guidelines established by the Ministry of Health in Mexico. All participants and healthy donors (controls) were informed and provided written consent to participate in the study.

### Patients

The number of participants varied across different study sections and these numbers are specified in each respective section. Patients diagnosed with LCL and DCL were residents of the State of Tabasco, an endemic area for *L. mexicana* in southeastern Mexico. All patients received anti-*Leishmania* treatment with pentavalent antimonials, as mandated by National Health Authorities. LCL patients presented active lesions with relatively few parasites. In contrast, DCL patients exhibited multiple nodular lesions characterized by heavily parasitized macrophages. The severity of DCL cases ranged from patients with nodules confined to the upper body to those with nodules affecting the entire body, limbs and oral and nasal mucosae. *Leishmania* parasites were detected in smears created from biopsy punches taken from the lesions, stained using Giemsa and subjected to histopathological analysis with hematoxylin and eosin (H&E) staining. For *in vitro* experiments, eosinophils from individuals with eosinophilia (n = 8) of unknown origin who sought care at the allergy ward in the Hospital General de México, were utilized. Only individuals with eosinophilia exceeding 6% were included in this study.

Peripheral blood eosinophils were quantified via flow cytometry, using FCS and SSC parameters, in patients with LCL (n = 30) and DCL (n = 4). Punch skin biopsies (Stiefel Laboratories, Inc., Coral Gables, FL, USA) were obtained from the lesions of LCL (n = 35) and DCL patients (n = 5), using 2% xylocaine as a local anesthesia. The biopsy specimens were fixed in 10% formalin and embedded in paraffin (Sigma-Aldrich, St. Louis, Mo, USA), cut into 4 μm thick sections, and stained with H&E, to evaluate eosinophil infiltration. Eosinophils were counted in eight images of each tissue, with a final area corresponding to 1 mm$^2$, using an Axio imager M1 microscope equipped with an MRc5 digital camera and Axiovision 4.8 software (Carl Zeiss, Industrielle Messtechnik GmbH, Oberkochen Germany).

### Detection of IgE and IgG in human serum against *Leishmania* mexicana antigens

We determined IgE and IgG levels against *L. mexicana* antigens through ELISA in LCL (n = 8) and DCL patients (n = 8), as well as in healthy controls (n = 8). Each sample was analyzed in triplicate. Briefly, 96-well EIA/RIA plates (Corning, NY, USA) were coated with 0.6 μg/100 μl of lysed promastigotes in each well and incubated for 1 hour. The solution was removed and the plates were blocked with 200 μl of 0.5% casein for 1 hour. After blocking, 100 μl of diluted (1:5) sera from leishmaniasis patients or healthy donors were added and incubated overnight

at 4°C. The plates were washed with PBS-Tween. For IgE detection, 100 μl of rabbit anti-human IgE (Dako, Agilent Technologies, Sta Clara, Ca, USA) (1:500) was added and incubated for 1 hour, followed by washing and the addition of 100 μl of biotin goat anti-rabbit IgG (1:8000) (Thermo Fisher Scientific, Waltham, Ma, USA) for 30 minutes. After washing, 100 μl of peroxidase-conjugated streptavidin (Zymed) (1:2500) was added and incubated for 30 min. For IgG detection, 100 μl of peroxidase-conjugated goat anti-human IgG (Zymed) antibody (1:4000) was added to the plates and incubated for 1 hour. Following washing, 100 μl of TMB (Tetramethylbenzidine) peroxidase substrate (Becton Dickinson, NJ, USA) was added for 20 minutes. The reaction was halted with 100 μl of 1 M phosphoric acid. Absorbance was measured at 450 nm using an ELISA microplate reader (BIO-TEK).

## Eosinophil isolation

Human eosinophils were obtained from peripheral blood samples of patients with LCL (n = 30) and DCL (n = 4). Cells were isolated through gradient centrifugation using Ficoll Hypaque (Sigma-Aldrich, St. Louis, Mo, USA). The blood was diluted 1:2 with sterile PBS pH 7.4 and layered over 20 mL Ficoll-hypaque (density $\rho$ = 1.077 g/mL) in a 50 mL conical tube. Centrifugation was performed at 600×g for 30 min at 20°C using a swinging bucket rotor. The subsequent steps involve removing and discarding the plasma. Mononuclear cells were located at the plasma:Ficoll interface, granulocytes and erythrocytes were in the pellet. The pellet was resuspended in 1X Red Blood Cell Lysis Solution, and the complete 50 mL conical tube was filled with lysis solution. Incubation occurred for 10 min on ice, followed by centrifugation at 300×g for 10 min at 20°C. The supernatant was carefully removed entirely. Cells were washed adding 50 mL of buffer and the cell number was determined. The cell suspension was centrifuged at 300×g for 10 min. The supernatant was aspirated completely, and the cells were resuspended in 50 μl of buffer per $5 \times 10^7$ cells. To this suspension, 50 μl of anti-CD16 conjugated microbeads (Miltenyi Biotec, Bergisch-Gladbach, Germany) were added per $5 \times 10^7$ cells and incubation occurred for 30 min at 4–8°C. Cells were washed by adding 1–2 mL of buffer per $10^7$ cells, followed by centrifugation at 300×g for 10 min. The supernatant was discarded, and the cells were resuspended to a final volume of up to $10^8$ cells in 500 μL of buffer. The suspension was then placed in a purification column, which had been previously washed with a buffer and positioned in the magnetic field of a MACS separator. Unlabeled cells, including eosinophils, were obtained by depletion of magnetically labeled cells. The total effluent containing the unlabeled cell fraction harbored the enriched eosinophil fraction. The purity and viability of eosinophils were >99%, as determined by Giemsa staining and by Trypan blue exclusion, respectively.

## *Leishmania mexicana* promastigotes culture

*L. mexicana* promastigotes were cultured in blood agar (NNN medium) overlaid with Schneider's *Drosophila* medium (Life Technologies), supplemented with 10% heat inactivated FBS (Biowest, Riverside, MO, USA) and 1% antibiotics (penicillin/streptomycin (Gibco, Life Technologies) at 28°C. Parasites were subcultured every 3–4 days and grown to a density of $1 \times 10^7$/ml.

## Cytokine measurement

The impact of *L. mexicana* on the production of IL-6, IL-8, and IL-13 by eosinophils from LCL (n = 7) and DCL (n = 7) patients, as well as healthy controls (n = 7) was analyzed as follows: $1 \times 10^6$ cells were incubated for 18 hours at 37°C and 5% $CO_2$ with $10 \times 10^6$ promastigotes in 1 ml of RPMI-1640 medium, supplemented with 10% heat-inactivated FBS. The cell-free

supernatants from the cultures were collected and the concentrations of the cytokines were determined using human IL-6 and IL-13 Quantikine ELISA Kits (R&D Systems Minneapolis, MN, USA). IL-8 was analyzed by a standard sandwich ELISA. Briefly, 96-well flat-bottom microtiter plates (Costar, Corning, NY, USA) were coated with 1 μg/ml unconjugated anti-cytokine capture antibodies (Purified mouse anti-human IL-8 monoclonal antibody Clone G265-5, PharMingen) diluted in 0.1 M Na2HPO4 (pH 9) and incubated overnight at 4°C. The plates were blocked with PBS (pH 7.4) supplemented with 0.5% casein dissolved in 0.1 N NaOH. Cell-free supernatants and recombinant hIL-8 standard (PharMingen) were incubated in RPMI-1640 medium supplemented with 10% FBS overnight at 4°C. Bound human IL-8 was detected using 0.5 μg/ml biotin-labeled detection antibodies (Biotinylated mouse anti-human IL-8 monoclonal antibody Clone G265-8, PharMingen) diluted in 1% BSA with 0.05% Tween 20 and incubated for 1 hour at room temperature. The plate was developed using AP-Streptavidin Conjugate (Life TechnologiesTM) and phosphatase substrate (0.005 mg/mL, Sigma-Aldrich). Absorbance was read at 405 nm and the IL-8 concentration of each sample was calculated by regression analysis based on a standard curve. The detection limit of this assay ranged from 15.6 to 1000 pg/ml. The cut-off point is determined from the lower detection limit, values below this limit considered negative. Unstimulated eosinophils were included in all cytokine production determinations; however, they were not included in the plot as they were below the cut-off point.

## Oxidative burst

The oxidative burst was measured by chemiluminescence immunoassay with eosinophils obtained from the same group of patients and controls as in the cytokine analysis. Briefly, $1x10^6$ eosinophils were incubated with $10x10^6$ promastigotes in 80 μl of RPMI 1640 medium, 210 μl of luminol (Sigma-Aldrich, St. Louis, Mo, USA) at a concentration of 1 mg/ml, and 60 μl AB+ serum-opsonized zymosan (12.5 mg/ml). Chemiluminescence was analyzed for 30 min (Luminoskan Labsystem, Finland) at 37°C with a 550 nm filter [12]. The values obtained were expressed in mVolt.

## Interaction of *Leishmania* promastigotes and blood eosinophils from healthy controls

The interaction between *L. mexicana* and eosinophils was analyzed using 5-carboxyfluorescein diacetate (CFDA) staining. *Leishmania* promastigotes ($10x10^6$) were washed twice with cold RPMI and stained with 5 μM CFDA (Sigma-Aldrich, St. Louis, Mo, USA) for 10 min in RPMI at 37°C. The parasites were washed three times with PBS at 26°C and incubated with $1x10^6$ eosinophils for 2 hours at 37°C, protected from light. They were then centrifuged and resuspended in 10–20 μl of PBS and analyzed with an epifluorescent microscope [13]. Some of the smears were stained with Giemsa.

## Transmission electron microscopy

Following careful washing with PBS, co-cultures of eosinophils ($1x10^6$) and promastigotes ($10x10^6$) were fixed in 2.5% glutaraldehyde diluted in 0.2 M cacodylate buffer, pH 7.4, for 2 hours, at 4°C. After three washes of 10 minutes each in 0.15 M cacodylate buffer, pH 7.4 at 4°C, the pellets of the cocultures were postfixed with 1% osmium tetroxide in 0.2 M cacodylate buffer, pH 7.4, for 30 min at 4°C. Dehydration was carried out in gradually increasing concentrations of ethanol, followed by transfer to absolute toluene, using two changes of 10 min each. Infiltration was done with araldite 6005-toluene solution (1:1) for 24 hours at room temperature and then in pure araldite 6005 (two changes) at 60°C for 2 hours. Specimens were

embedded in pure araldite 6005 at 60°C, for 36 hours. Ultrathin sections were obtained with a diamond knife, contrasted with uranyl acetate and lead citrate, and examined using an Electron Microscope (EM) 109 (Carl Zeiss, Germany). All reagents used were from Electron Microscopy Science, Hatfield, PA, USA.

## Immunocytochemistry for detection of eosinophil major basic protein (MBP)

For the immunocytochemical analysis, $1x\,10^6$ purified eosinophils were incubated in a chamber slide (Nunc Lab-Tek, Sigma-Aldrich, St. Louis, Mo, USA) for 2 hours at 37°C and 5% $CO_2$ with $10x10^6$ *L. mexicana* promastigotes in 1 ml of RPMI-1640 medium supplemented with 10% heat-inactivated FBS. After incubation, they were washed with PBS and fixed with 2% paraformaldehyde (Sigma-Aldrich, St. Louis, Mo, USA). Nonspecific antigen sites were blocked with 2% bovine serum albumin, dissolved in Tris-HCl, pH 7.6. Thereafter, they were stained with mouse anti-human MBP at a dilution of 1:100 (Santa Cruz Biotechnology, Ca, USA) for 30 min. After washing, eosinophils were incubated with the secondary antibody, biotin goat anti-mouse at a dilution of 1:100 (Invitrogen, Waltham Ma. USA) for 30 min, washed and incubated with streptavidin AP (Alkaline Phosphatase) at a dilution of 1:100 (Thermo Fisher Scientific, Waltham Ma. USA) for 30 min. After washing, they were counterstained with hematoxylin and mounted with resin. Additionally, the incubation of eosinophils with promastigotes was also stained with DAPI in separate samples (Sigma-Aldrich, St. Louis M. USA) to verify the formation of DNA nets.

## Statistical analysis

Comparisons between experimental groups and controls were performed using Mann–Whitney U-test. A value of $p < 0.05$ was considered statistically significant, using Prism 8 for Windows.

## Results

### Eosinophils in peripheral blood and lesions of LCL and DCL patients

Blood samples were collected from 30 LCL patients (19 males and 11 females), with a mean age of 28 years (range: 16–78 years) and disease durations ranging from 1.5 to 18 months (mean: 16 ± 3.98). Four DCL patients (three males and one female) with a mean age of 37 years (range: 20–45 years) and disease durations ranging from 40–120 months (mean: 67 ± 18.14 months) were also included. Eosinophils were quantified in peripheral blood using flow cytometry. The mean percentage of eosinophils in LCL patients was 11.1 ± 1.51 with 60% of LCL patients exhibiting eosinophilia ranging from 5% to 32% (Fig 1A). There were no significant correlations between the number of eosinophils in peripheral blood and disease duration, age or gender (data not shown). Notably, all female patients had a shorter disease duration compared to male patients (Fig 1B), with females showing an average disease evolution of 6.5 ± 1.45 months, while males had an average of 21.4 ± 5.95 months. All DCL patients exhibited blood eosinophilia, with a mean percentage of eosinophils in the blood of 16 ± 5.8%, ranging from 7% to 34% (Fig 1A). It is noteworthy that all DCL and few LCL patients with chronic infections lasting over two years had blood eosinophilia (Fig 1B).

### Eosinophils in lesions of LCL and DCL patients

Skin biopsies were obtained from 35 LCL patients and 5 DCL patients. When analyzing skin lesions, significant differences were observed between eosinophil numbers of both clinical

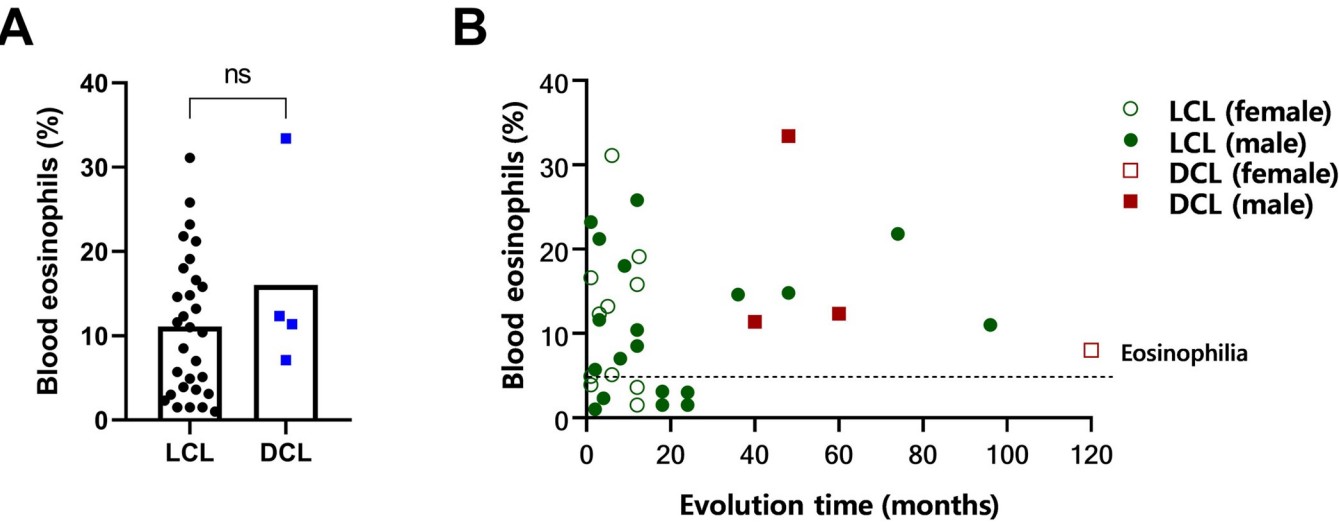

**Fig 1. Eosinophils in peripheral blood of patients with cutaneous leishmaniasis.** A. Percentage of blood eosinophils from patients with LCL (black circles, n = 30) and DCL (blue squares, n = 4). B. Blood eosinophils and disease evolution time according to disease form and gender are shown: LCL male patients are indicated with green filled circles; DCL male patients are indicated by filled red squares. LCL female patients are indicated by empty green circles; DCL female patients are indicated with empty red squares). The horizontal line represents the mean, and each symbol represents a patient (female or male). Bars represent the mean ± SEM. ns: non-significant.

forms (Fig 2A). Eosinophils were observed in skin lesions of all 5 DCL patients with a mean of 14 ± 1.15 eosinophils/mm$^2$ (ranging from 11 to 18). In contrast, only 20% of LCL patients had eosinophils in their tissues, with a mean of 3 ± 1 eosinophils/mm$^2$ (ranging from 0 and 18). When comparing eosinophils in tissues between LCL and DCL patients, a significant difference was found between the groups, with significantly higher numbers in DCL patients (Fig 2A). However, when specifically comparing LCL patients with tissue eosinophils to all DCL patients, no significant differences were found between the two groups of patients. LCL patients with tissue eosinophils showed an average of 13 ± 1.6 eosinophils/mm$^2$ (ranging between 5 and 18), while DCL patients showed an average of 14 ± 1.1 eosinophils/mm$^2$ (ranging between 11 and 17). When comparing eosinophils in tissues between female and male of LCL and DCL patients, a significant difference was found only between males (Fig 2B). Histopathological analysis of infected tissues revealed that DCL patients had loss of tissue integrity and a higher parasite load, compared to those of LCL patients (Fig 2. bottom panel).

### Anti-*Leishmania* IgE and IgG in sera from LCL and DCL patients

Serum levels of IgE and IgG were measured in LCL (n = 8) and DCL (n = 8) patients to determine the serological response to *L. mexicana*. Both antibodies were identified in all patients, with no significant differences in IgE levels between DCL (1.43 ± 0.18) and LCL patients (0.95 ± 0.19). However, both patient groups produced significantly more IgE than controls (0.167 ± 0.068) (Fig 3A). When comparing IgG production, DCL patients (1.73 ± 0.05) produced significantly more than LCL patients (1.31 ± 0.11), and both patient groups produced more IgG than controls (0.293 ± 0.074) (Fig 3B).

### Cytokine (IL-6, IL-13, and IL-8) production and oxidative burst by EOSINOPHILS of LCL and DCL patients incubated with *L. mexicana* promastigotes

Our data indicate that IL-6 and IL-13 production by eosinophils from patients with DCL (n = 7) was significantly higher, compared to those from LCL patients (n = 7) or healthy

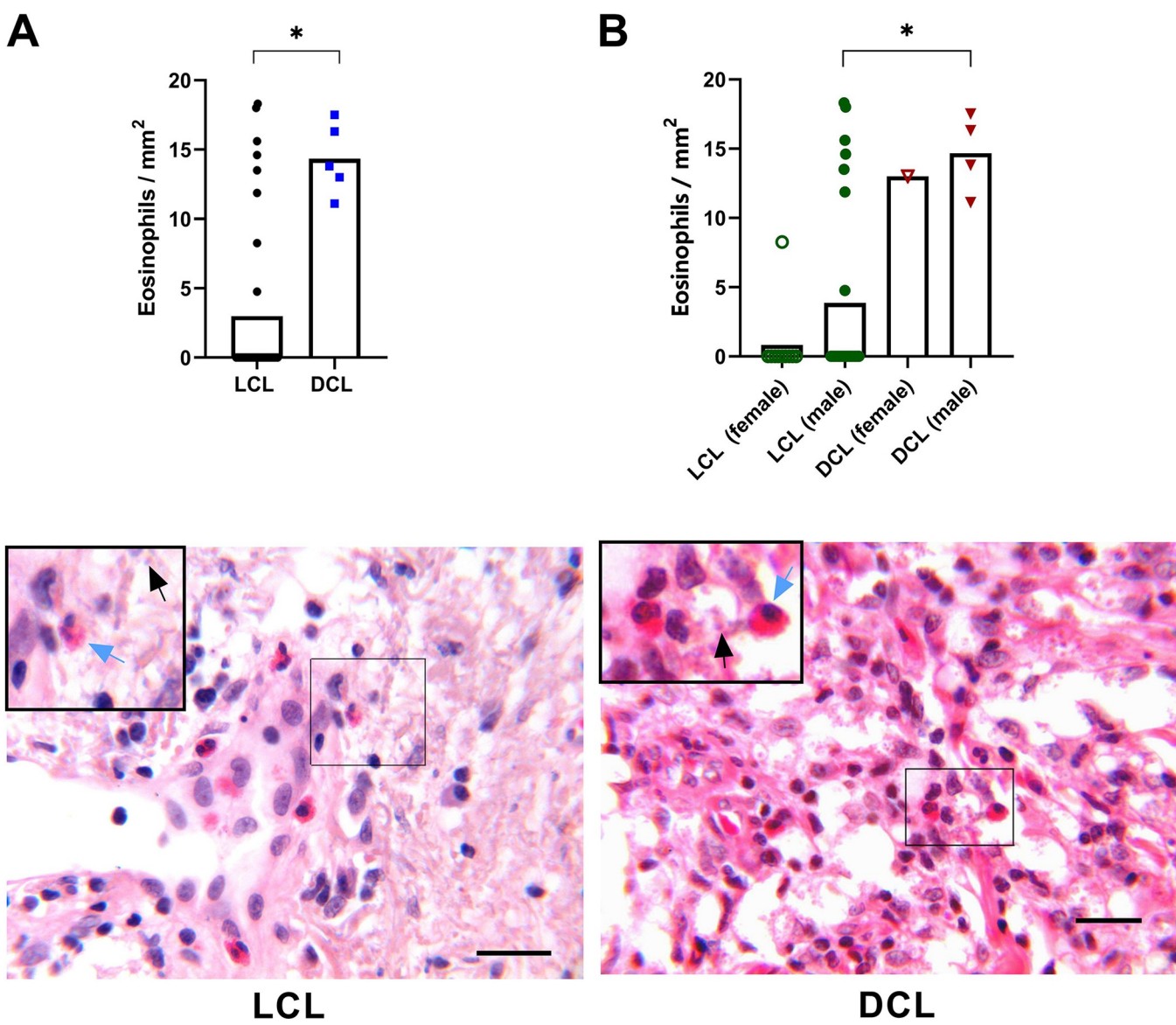

**Fig 2. Eosinophils in lesions of patients with cutaneous leishmaniasis.** A. Eosinophils in the tissue were quantitated and expressed as eosinophils/mm$^2$. LCL patients (black circles, n = 35) and DCL patients (blue squares, n = 5). B. Eosinophils/mm$^2$ in tissue lesions according to disease form and gender. Male LCL patients (n = 25, filled green circles) and female LCL patients (n = 10, empty green circles); Male DCL patients (n = 4, filled red triangles) and female (n = 1, empty red triangle). Bottom panel: Representative H&E staining of biopsies taken from LCL and DCL patients. Black arrows indicate *Leishmania* amastigotes and blue arrows indicate eosinophils. Scale bar = 20 μm. Bars represent the mean ± SEM, and asterisks represent statistically significant differences (P < 0.05).

control subjects (n = 7), when incubated with *L. mexicana* promastigotes. IL-6 production by DCL eosinophils (45 ± 13.2 pg/mL) was significantly higher compared to LCL patients (7.4 ± 1 pg/mL). Control subjects did not exhibit IL-6 production (Fig 4A).

IL-8 production by eosinophils of DCL patients (45 ± 2.6 pg/mL) was also significantly higher compared to LCL patients (29 ± 2 pg/mL) and control subjects (40 ± 1.7 pg/mL). Interestingly, eosinophils of control subjects produced nearly the same amount of IL-8 as those of DCL patients) (Fig 4B).

In the case of IL-13 production, eosinophils of DCL patients showed a significantly higher production (1000 ± 11.5 pg/mL) compared to LCL patients and controls (50 ± 2.8 pg/mL) (Fig

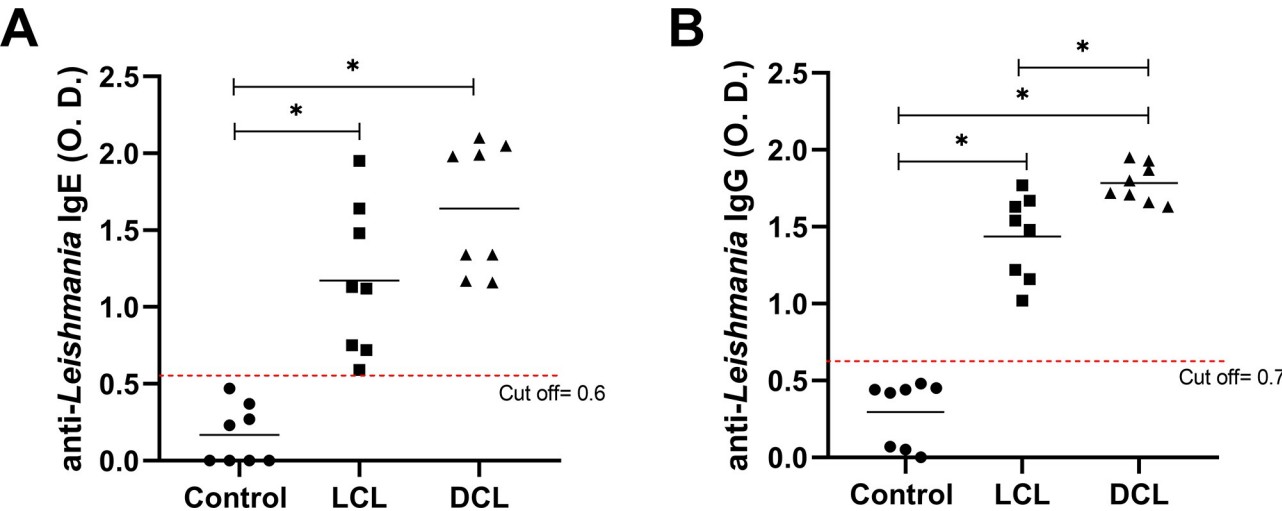

**Fig 3. Anti-*Leishmania* IgE and IgG in sera from patients with cutaneous leishmaniasis.** A. IgE levels. B. IgG levels. Each symbol represents a patient, and the horizontal line represents the mean of each group. LCL (n = 8) and DCL (n = 8) patients, healthy controls (n = 8). *All symbols represent statistically significant differences (P < 0.05).

4C). These findings demonstrate that eosinophils from DCL patients produce significantly more IL-6, IL-8, and IL-13 when exposed to *L. mexicana* promastigotes compared to those of LCL patients. Unstimulated eosinophils did not exhibit IL-6, IL-8, and IL-13 cytokine production and were below the ELISA cut-off point.

The analysis of the oxidative burst in eosinophils of LCL (n = 7), DCL (n = 7) and healthy subjects (n = 7) co-incubated with *L. mexicana* promastigotes showed that the parasite enhances the magnitude of the oxidative burst in eosinophils of both groups of patients to an equal extent, which was significantly higher compared to eosinophils from healthy controls (Fig 4D). However, no significant differences between both clinical forms.

## Phagocytosis and lysis of *Leishmania* mexicana promastigotes by eosinophils

Microscopic analysis revealed that purified eosinophils do not release granules in the absence of parasites (Fig 5A). When co-incubated with *L. mexicana*, eosinophils exhibited the ability to damage extracellular promastigotes by releasing cytoplasmic granules. This damage resulted in the alteration of parasite morphology and the acquisition of a deformed phenotype (Fig 5B). Giemsa staining further demonstrated that damage occurred when *Leishmania* was in close contact with the eosinophils, as well as when parasites were surrounded by clouds of massively released granules (Fig 5B). Staining with CFDA revealed that *L. mexicana* promastigotes began to form clusters in the culture medium (Fig 5C). After incubation with eosinophils, the promastigotes are phagocytosed and transformed into intracellular parasites, ultimately leading to their degradation (Fig 5D). The percentage of eosinophil degranulation was significantly higher in eosinophils co-incubated with *Leishmania* (Fig 5E).

## Eosinophils release cytoplasmic "degranulation sacs" with granules in response to *Leishmania*

Co-incubation of eosinophils with *L. mexicana* revealed that, in addition to diffusely releasing cytoplasmic granules near parasites, eosinophils also release cytoplasmic "degranulation sacs" containing granules with electron-dense characteristics, surrounded by a membrane (Fig 6).

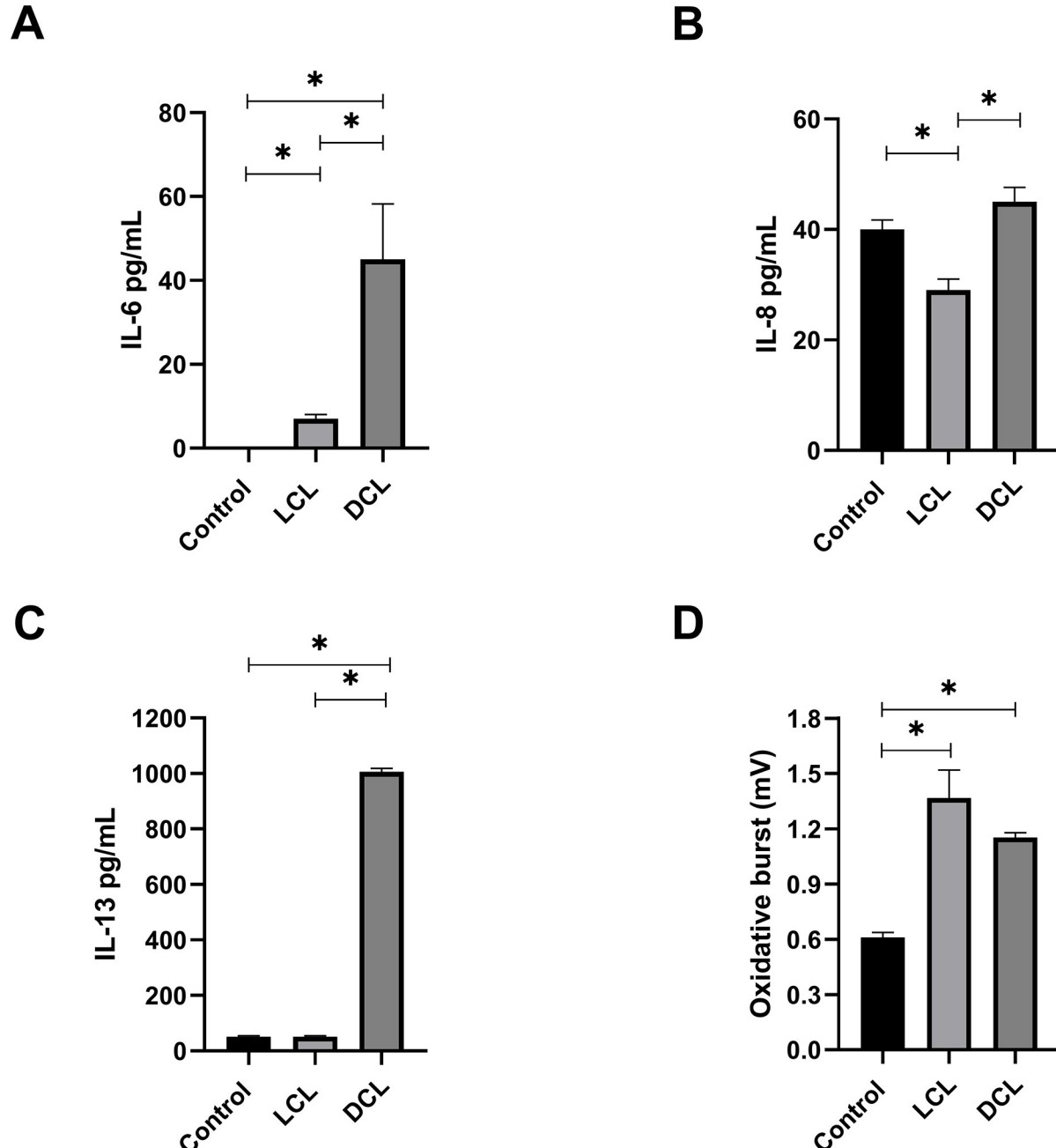

**Fig 4. Cytokine production and oxidative burst were analyzed in eosinophils from patients with LCL and DCL.** Eosinophils from LCL and DCL patients and from healthy controls were incubated with *L. mexicana* promastigotes. A. IL-6 production. B. IL-8 production. C. IL-13 production. D. ROS production measured by luminol reaction (in mV). Bars represent the mean ± SEM ($n = 7$ for each group of patients and healthy control subjects). All symbols represent statistically significant differences ($P < 0.05$).

Subsequently, as the cell membrane was lost, individual granules came into close contact with the parasites, leading to damage (Fig 6A and 6B). Electron microscopy analysis further illustrated that eosinophils could shed substantial portions of their cytoplasm, including granules, in the form of "degranulation sacs" onto the parasite. This phenomenon was evidenced by the presence of submembrane microtubules and caused significant damage to the parasites. Signs

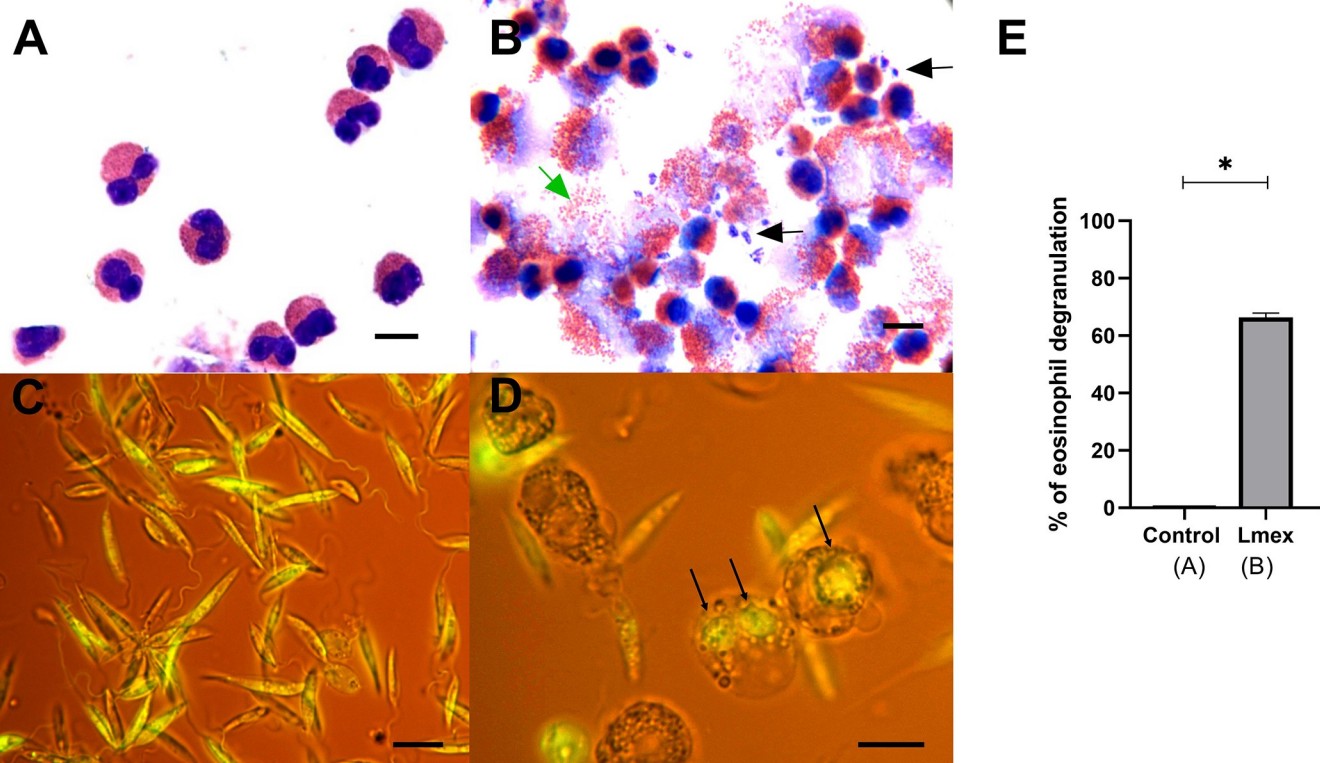

**Fig 5. Phagocytosis and lysis of *Leishmania mexicana* by eosinophils.** A. Purified eosinophils in the absence of parasites. B. Eosinophils incubated with *L. mexicana* promastigotes (Giemsa stained). Close contact between parasites and cells leads to abundant degranulation (green arrow) and damaged parasites (black arrows). C. *Leishmania* promastigotes stained with CFDA. D. Eosinophils co- incubated with *Leishmania* promastigotes show phagocytosed parasites (black arrow). Scale bar = 20 μm. E. Percentage of eosinophil degranulation with or without *L. mexicana* (n = 3). Bars represent the mean ± SD, and the asterisk represents statistically significant differences (P <0.05).

of parasite damage included diffuse osmophilic areas, indicating damage to intracellular membranes (Fig 6C and 6D).

## Extracellular traps are induced by *L. mexicana*

Another leishmanicidal mechanism observed *in vitro* was the release of extracellular traps (EETs) by eosinophils, which entrapped the parasites and brought them into contact with released granules such as MPB (7A). Under resting conditions, only eosinophil nuclear DNA was stained blue (7B). However, after coincubation with *Leishmania*, eosinophils released nets containing DNA, as indicated by blue DAPI staining, which binds DNA (Fig 7C). These DNA nets effectively trapped the parasites (Fig 7D).

## Discussion

Eosinophils, constituting 1–5% of human blood leukocytes, play a multifunctional role in defending against various infections, including parasites, viruses, bacteria, and fungi [4,5,14]. However, the understanding of eosinophils' role in patients infected with *L. mexicana*, presenting varying disease severity, such as localized (LCL) or diffuse (DCL) cutaneous leishmaniasis, remains limited.

 In our study, we observed that all DCL patients, and a subset of LCL patients with prolonged disease evolution exhibited both blood and tissue eosinophilia. Blood eosinophilia has

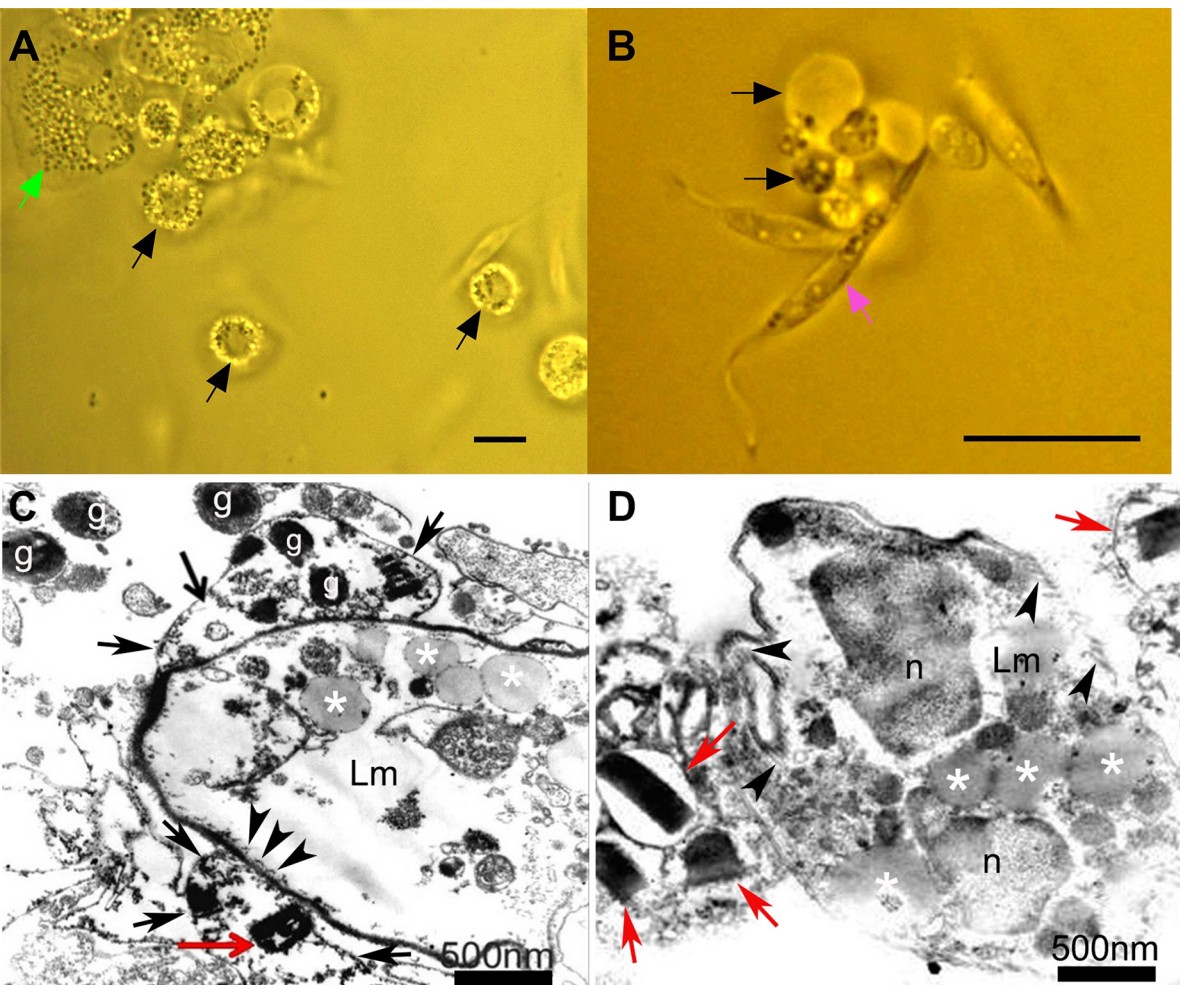

**Fig 6. Release of cytoplasmic "degranulation sacs" by eosinophils co-incubated with parasites.** A. Eosinophil (green arrow) and release of cytoplasmic "degranulation sacs". B. Degranulation sacs (black arrows) and parasites (pink arrow). C and D. Ultramicroscopic images showing released specific or secondary granules (g) or within cytoplasmic pockets limited by an intact membrane (black arrows). These "degranulation sacs" were in close position to distorted *L. mexicana* (Lm). Parasites, identified by their submembranous microtubule cytoskeleton (arrowheads) show signs of damage as evidenced by osmiophilic areas (white asterisk), nuclear fragmentation (n), and a clear or aqueous cytoplasm denoting an extensive cytoplasmic vacuolization. (D) Some eosinophil specific granules still retain the characteristic electron-dense central crystalline core, surrounded by a clearer peripheral matrix (red arrows). Light microscopy scale bar = 20 μm. TEM scale bar = 500 nm.

previously been reported in *L. mexicana*-infected DCL patients and recently cured LCL patients [15], as well as in patients infected with high numbers of *L. donovani* parasites [16]. Mouse models of leishmaniasis have also demonstrated that the eosinophil responses depend on mouse strains, *Leishmania* species, and the phase of infection [10,17,18]. Elevated eosinophil numbers, persisting throughout the infection, have been described in susceptible mouse strains infected with *L. mexicana* and *L. amazonensis*, which correlated with enhanced tissue damage and high numbers of intracellular amastigotes [6,10].

Eosinophils employ various mechanisms to control pathogens, including oxidative bursts, phagocytosis, and the release of eosinophil extracellular traps (EETs). EETs contain histones, eosinophil cationic proteins, and cytotoxic granules such as MBP [19,20] within "degranulation sacs", which are released through a single fusion pore [21]. Furthermore, the shedding of clusters of free extracellular eosinophil granules, as well as of plasma membrane-enveloped

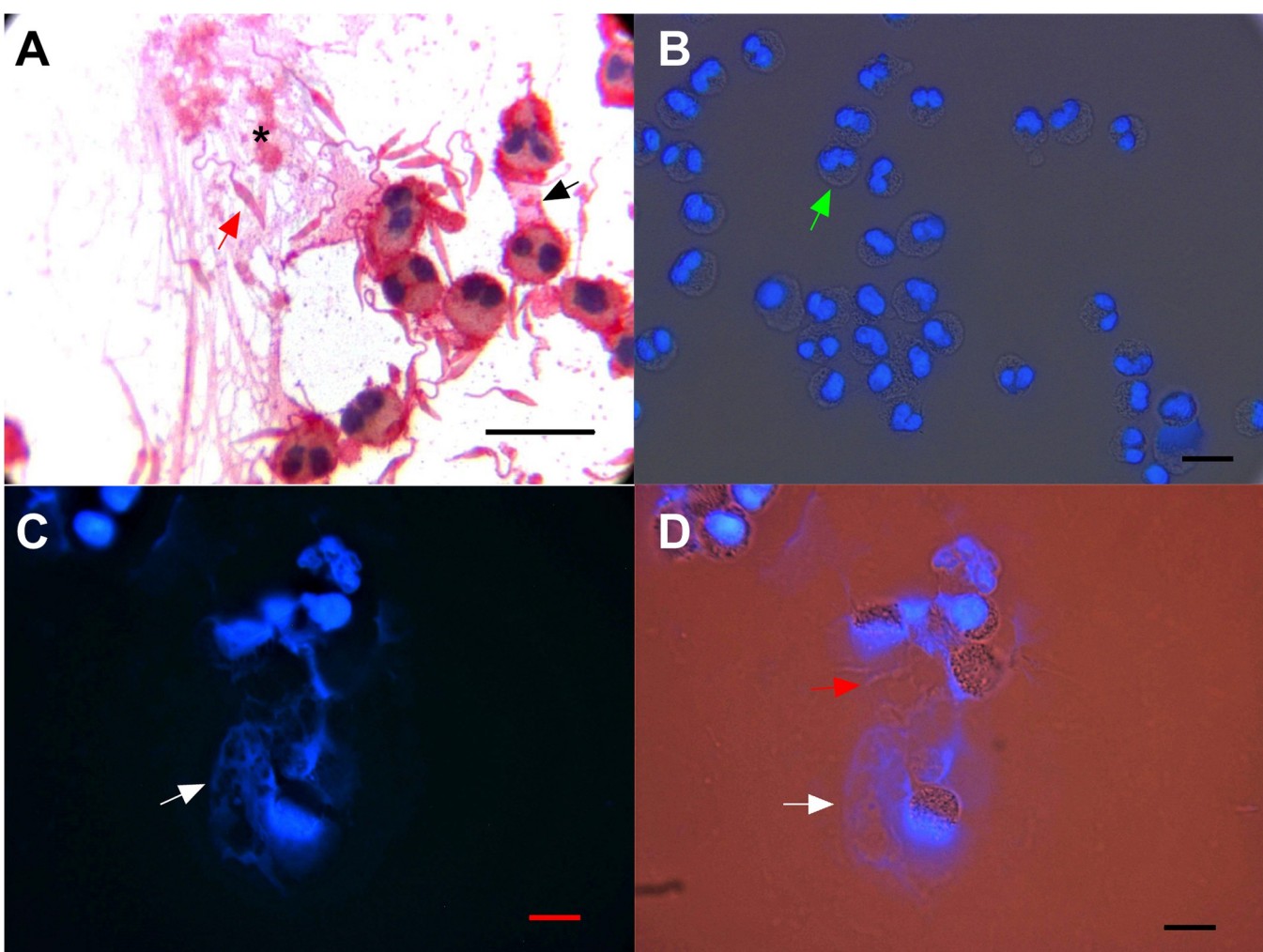

**Fig 7. Extracellular eosinophil traps induced by co-incubation with *L. mexicana*.** A) Extracellular traps (black asterisk) containing MBP-positive granules (black arrow) and parasites within the nets (red arrow). B) Purified eosinophils show their nuclei stained blue with DAPI (green arrow). No nets are formed in the absence of *Leishmania*. C) Eosinophils co-incubated with *L. mexicana* promastigotes show the formation of extracellular DNA traps (white arrow). D) The same image (C) was analyzed with light and fluorescence microscopy, showing the parasites (red arrow) in contact with nets (white arrow). Scale bar = 20 μm.

structures containing cytoplasmic granules, has been reported. After membrane lysis, the granules are released into the extracellular space, where they function as free granules [22].

Our findings demonstrate that eosinophils phagocytize *L. mexicana* promastigotes *in vitro* (Supplementary 1), which is accompanied by various forms of granule exocytosis that inflict damage to the parasite, as confirmed by electron microscopy and Giemsa staining. The contact between the parasites and eosinophils induced the release of free granules and the extrusion of membrane bound "degranulation sacs" containing numerous granules, which are released near the parasites, leading to damage (Supplementary 2). This phenomenon resembles a piecemeal degranulation process [21,22]. Furthermore, eosinophils released EETs containing MBP in response to *L. mexicana* parasites. Our data align with previous reports of eosinophils attaching to and phagocytizing *Leishmania mexicana amazonensis* parasites, resulting in oxidative bursts and the release of extracellular traps containing membrane-bound clusters of granules, as well as free granules, both of which potentially contribute to parasite elimination [23–25]. Although our observations were conducted *in vitro*, this phenomenon may be expected to occur in infected tissues, thus contributing to tissue damage [26,27].

Eosinophil cytokines modulate the immune response, but their involvement in defining the virulence of cutaneous leishmaniasis is not well understood. Leishmaniasis, a disease with divergent outcomes based on the balance of pro- and anti-inflammatory cytokines, is regulated by cytokine-producing cells. Th1 cytokines support parasite control, while a Th2 response is associated with susceptibility [5,14]. Given that the immune response in DCL patients cannot effectively control the extensive spread of parasites, we were interested in examining whether eosinophil responses to *L. mexicana* differ between DCL and LCL patients, potentially favoring disease progression.

Our data now show that eosinophils from DCL patients secrete significantly higher levels of IL-6, IL-8 and IL-13 in response to *L. mexicana* promastigotes, compared to eosinophils from LCL patients. This differential response between DCL and LCL eosinophils, may be related to the genetic background of patients, as previous studies have reported differential expression of inflammatory genes in NK cells from DCL and LCL patients following TLR stimulation with LPG, with LCL NK cells expressing more genes associated to inflammatory responses compared to DCL NK cells [28].

Previous studies have reported the presence of IL-6 during the active phases of infection by several *Leishmania* species [29–31]. IL-6 along with IL-1 and IL-23 regulates Th17 cell differentiation [31]. The Th17 response promotes the chemotaxis of neutrophils through the induction of IL-8 on fibroblasts, epithelial, and endothelial cells [32]. In experimental cutaneous leishmaniasis, the Th17 response has been associated with an increase in pathological conditions, primarily due to the neutrophil-mediated promotion of Th17 response [33].

Early recruitment of neutrophils to the infection site mediated by IL-8 also leads to their activation and enhanced phagocytosis of the parasite. Neutrophils not only harbor the parasite, shielding it from the deleterious effects of complement, but they have also been described to aid in parasite spread through "silent" phagocytosis that occurs after apoptotic cells containing the parasites are phagocytosed by non-activated macrophages ("Trojan horse" strategy) [34–36]. Our results of enhanced production of this chemokine are in line with the literature, where infections by *L. infantum*, and *L. major* have all been shown to enhance the production of IL-8 [37–39]. Additionally, aside from disease modulation resulting from enhanced IL-8 production, polymorphisms in IL-8 (-251A/T) or IL-6 (174G/C) genes should also be considered in future studies, as they have been shown to influence the outcome of mucosal leishmaniasis [40–42]. This aspect remains to be analyzed in cutaneous leishmaniasis caused by *L. mexicana*.

In visceral leishmaniasis (VL) caused by *L. infantum*, higher IL-6 and IL-17 levels were found in male patients, as compared to females, which correlated with enhanced disease severity in males [43,44]. We now show that IL-6 production by eosinophils was higher in DCL patients as compared to LCL patients, suggesting that IL-6 plays a role in disease severity in cutaneous leishmaniasis. Furthermore, our finding of elevated production of IL-13 by eosinophils of DCL patients is another factor that might contribute to enhanced disease severity. IL-13 activates eosinophils and promotes the development of alternatively activated macrophages, which have been shown to promote disease progression in cutaneous leishmaniasis [14,31,37]. This cytokine is functionally redundant with IL-4 and inhibits the production of inflammatory cytokines, thereby interfering with resistance to *Leishmania* [45]. Furthermore, IL-13 can promote the isotype switching of B cells to produce IgE and IgG4 in humans [46]. Our observations reveal that both DCL and LCL patients have elevated anti-*Leishmania* IgE and IgG antibodies. Although no differences were observed in IgE production between both clinical forms, it is noteworthy that both patient groups tested positive for IgE, contrasting with other reports of IgE in *L. amazonensis* infections, where only 40% of patients were positive for this antibody isotype [47]. However, we found that DCL patients had significantly higher levels of

IgG, compared to LCL patients. This is noteworthy, as higher amounts of IgG in leishmaniasis have been associated with susceptibility, as IgG induces the production of regulatory IL-10 in monocytes, contributing to the inhibition of cellular responses in chronic DCL patients [48].

In summary, these three cytokines have been associated with disease susceptibility, and our data align with the literature, showing their enhanced production by eosinophils from DCL patients under *in vitro* conditions. It is possible the higher eosinophil numbers in the infiltrated inflammatory areas of DCL patients contribute to the production of these cytokines, further exacerbating disease severity. However, this hypothesis remains to be investigated in the context of cutaneous leishmaniasis caused by *L. mexicana*.

We found that all female patients had a shorter disease duration compared to male patients. This could be due to differences in the immune system between females and males, that have already been well documented [49]. These differences possibly affect disease susceptibility to leishmaniasis. In male mice infected with *L. major*, parasite load was shown to be related to eosinophils. A significant correlation between elevated eosinophil infiltration and increased parasite loads was only found in male mice infected with *L. major*, suggesting that eosinophils seem to be related to disease severity in males [50]. Thus, circulating sex hormones, and possibly additional parasite factors, affect the immune response and play a role in the disease outcome [51,52]. The influence of sex hormones on *Leishmania* parasites was shown in experiments where *Leishmania mexicana* was incubated with dihydrotestosterone, which led to their enhanced growth and infectivity and may be one of the possible factors associated with male susceptibility to *Leishmania* [53]. It remains to be analyzed whether men with a longer disease course, are also more likely to develop eosinophilia.

The results obtained shed new light on the apparent dual role of eosinophils in leishmaniasis: they can inflict crucial damage to the parasite showing a healing effect, whereas they can also cause worsening of the disease by producing cytokines such as IL-6, IL-8 and IL-13 upon contact with the parasite, associating eosinophils with disease severity and pathological conditions.

## Conclusions

Eosinophils employ diverse effector mechanisms when co-incubated with *Leishmania mexicana*, including the exocytosis of granules onto the parasite, the release of membrane bound "degranulation sacs", and the formation of EETs, all of which contribute to parasite damage. However, despite presenting eosinophilia, DCL patients exhibit a high parasite load and uncontrolled disease progression. Given that eosinophils from DCL patients also release significantly higher amounts of cytokines IL-6, IL-13, and the chemokine IL-8, compared to LCL patients, these mediators may contribute to the disease progression observed in DCL patients. Our data provide additional insights into the innate defense mechanisms exerted by eosinophils against *Leishmania mexicana*. To further clarify the biological significance of eosinophils in leishmaniasis it would be interesting to study other effector mechanisms, including signaling pathways, signal transductions, and polymorphisms in the eosinophils of these patients, in order to enhance our understanding of eosinophils in this disease.

## Supporting information

**S1 Fig. Phagocytosed parasites by eosinophils.** A) Parasites stained with CFDA (green color) phagocytosed by eosinophils are observed inside these cells. Black arrows show phagocytized parasites. B) Remains of degraded parasites (black arrows) are observed already inside the eosinophils (green color granules). The co-incubation ratio was 1:10 for 2 hours. Scale

bar = 20 μm.
(TIF)

**S2 Fig. *Leishmania mexicana* damaged by eosinophil contact.** A) Normal shape and size of a viable *Leishmania* promastigote. B) Morphological changes in parasite size, damage to membranes, formation of small vacuoles within the parasite, and loss of flagellum were observed in parasites co-incubated with eosinophils in a 1:10 ratio for 1 hour. Black arrows show damaged parasites. Scale bar = 20 μm.
(TIF)

## Author Contributions

**Conceptualization:** Norma Salaiza-Suazo, Roxana Porcel-Aranibar, José Delgado-Domínguez, Ingeborg Becker.

**Formal analysis:** Isabel Cristina Cañeda-Guzmán, Jaime Zamora-Chimal, José Delgado-Domínguez, Héctor A. Rodríguez-Martínez.

**Funding acquisition:** Ingeborg Becker.

**Investigation:** Norma Salaiza-Suazo, Isabel Cristina Cañeda-Guzmán, Adriana Ruiz-Remigio, Jaime Zamora-Chimal, Rocely Cervantes-Sarabia, Georgina Carrada-Figueroa, Baldomero Sánchez-Barragán, Victor Javier Leal-Ascencio, Armando Pérez-Torres.

**Methodology:** Norma Salaiza-Suazo, Roxana Porcel-Aranibar, Adriana Ruiz-Remigio, José Delgado-Domínguez, Armando Pérez-Torres.

**Project administration:** Ingeborg Becker.

**Resources:** Georgina Carrada-Figueroa, Baldomero Sánchez-Barragán, Victor Javier Leal-Ascencio.

**Supervision:** Ingeborg Becker.

**Visualization:** Norma Salaiza-Suazo, Isabel Cristina Cañeda-Guzmán, Jaime Zamora-Chimal, Rocely Cervantes-Sarabia, Georgina Carrada-Figueroa, Victor Javier Leal-Ascencio, Armando Pérez-Torres.

**Writing – original draft:** Norma Salaiza-Suazo, Roxana Porcel-Aranibar.

**Writing – review & editing:** Isabel Cristina Cañeda-Guzmán, Jaime Zamora-Chimal, Armando Pérez-Torres, Ingeborg Becker.

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
