## [Decision Letter · Decision Letter 0]

11 Apr 2023

PONE-D-23-05814Eosinophils of patients with localized and diffuse cutaneous leishmaniasis: differential response to Leishmania mexicana, with insights into mechanisms of damage inflicted upon the parasites by eosinophilsPLOS ONE

Dear Dr. Becker,

Thank you for submitting your manuscript to PLOS ONE. After careful consideration, we feel that it has merit but does not fully meet PLOS ONE’s publication criteria as it currently stands. Therefore, we invite you to submit a revised version of the manuscript that addresses the points raised during the review process.

We look forward to receiving your revised manuscript.

Kind regards,

Vyacheslav Yurchenko, Ph.D.

Academic Editor

PLOS ONE

Journal Requirements:

Additional Editor Comments (if provided):

The work "Eosinophils of patients with localized and diffuse cutaneous leishmaniasis: differential response to Leishmania mexicana, with insights into mechanisms of damage inflicted upon the parasites by eosinophils" was reviewed by two independent reviewers. While both felt that this work is important and timely, they have identified areas, which should be improved before reconsideration. Therefore, I request a major revision. All the Reviewers' concerns should be addressed. 

Reviewers' comments:

Reviewer's Responses to Questions

**Comments to the Author**

1. Is the manuscript technically sound, and do the data support the conclusions?

Reviewer #1: Partly

Reviewer #2: Partly

2. Has the statistical analysis been performed appropriately and rigorously? 

Reviewer #1: I Don't Know

Reviewer #2: N/A

3. Have the authors made all data underlying the findings in their manuscript fully available?

Reviewer #1: Yes

Reviewer #2: No

4. Is the manuscript presented in an intelligible fashion and written in standard English?

Reviewer #1: Yes

Reviewer #2: Yes

5. Review Comments to the Author

Reviewer #1: The article “Eosinophils of patients with localized and diffuse cutaneous leishmaniasis: differential response to Leishmania mexicana, with insights into mechanisms of damage inflicted upon the parasites by eosinophils “ by Norma Salaiza-Suazo et al., describe in several experiments the presence and response of eosinophils in LCL and DCL. Eosinophils have been neglected in the leishmaniasis contexts, probably because their significance is overshadowed by the importance of macrophages and T cells. This study address a relevant gap in the knowledge. However results here shown are rather descriptive than conclusive.

Major comments:

Methods section is in many points poorly described:

-Need more specification on the eosinophil isolation: How long was centrifuged? How was done the RBC lysis? How many washes/time/g-force?

-Leishmania was cultured in the absence of antibiotics?

-Also in-house method for IL-8 measurement lacks important information: which capture/detection antibody was used? Please describe the protocol.

-How was the cytometry done? What was the antibodies used and gating strategy applied?

-How was the cut-off in ELISA calculated?

-Why is the luminescence in oxidative burst measured as Mmvolt?

Line 309 – 310 “These findings demonstrate that eosinophils from patients with

DCL produce significantly more IL-6, IL-8, and IL-13 when exposed to L. mexicana

promastigotes, compared to those of LCL patients.” Cytokine measurement should have been done with unstimulated eosinophils. This is important to be sure that the cytokine production is not due to a pre-activated status. How could the authors be sure the cytokines produced were induced by parasites rather than being already activated?

Line 318-330 – To assume that eosinophils are being harmful against leishmania, the authors should show quantitatively increased death of promastigotes while in contact with Leishmania. They could use cellular death markers (Propidium iodide for example) to quantify the number of dead promastigotes in contact with eosinophil.

Lines 407-409 – It is not correct to associate IL-6 with TH2 responses. IL-6 is mainly produced by myeloid cells (mostly macrophages) and this cytokine is increased during VL together with a TH2 response, independently. IL-6 is required to a TH17 response and whether their production by eosinophils is important/significant during cutaneous leishmaniasis is yet to be shown.

Line 435: The authors says that “it is noteworthy that 70% LCL and 90% DCL were positive for IgE”, although figure 3 clearly shows (as well as is described in the text) that all patients were IgG and IgE positive. Please revise this excerpt and figure.

Lines 442 – 444 – authors must be careful to assume “an enhanced production of cytokines” in DCL patients, since they only measured eosinophil derived cytokines and this does not mean that these cytokines are in fact increased in this patients or even related to susceptibility (for example, IFN-g is associated with resistance and it is also increased during VL). The production of these cytokines by eosinophils compared to other cells (macrophages and monocytes) might be insignificant in natural conditions.

Figure 5 - Authors claim to demonstrate phagocitozed leishmania by eosinophil but the image has bad quality. Do authors have clear evidence of phagocytosis by eosinophils (stained slides would be enough).

Authors should discuss why an innate immune cell would respond differently in DCL and LCL. Which possible mechanisms would allow eos from DCL to be more prone to respond to promastigotes than LCL?

Line 449: “their release of the cytokines IL-6, IL-8 and IL-13 promotes disease severity”: the data here shown do not support this statement.

Reviewer #2: The authors report on an important and up-to-now largely neglected aspect of a cellular repsonse to Leishmania infection. This study certainly will be of great interest.

Major concerns:

- The statistical method employed should be detailled in the respective Fig. legends. The authors might want to use multivariate testing.

- The authors should demonstrate a by FACS that their MACS-based enrichment of eosinophils worked.

- The Fig. 5 - 7 should be analyzed using a readout that allows statistical analysis. Otherwise these findings seem very anectodal.

6. PLOS authors have the option to publish the peer review history of their article (what does this mean?). If published, this will include your full peer review and any attached files.

Reviewer #1: No

Reviewer #2: No

---

## [Author Response · Author response to Decision Letter 0]

28 Sep 2023

Reviewer #1: The article “Eosinophils of patients with localized and diffuse cutaneous leishmaniasis: differential response to Leishmania mexicana, with insights into mechanisms of damage inflicted upon the parasites by eosinophils “ by Norma Salaiza-Suazo et al., describe in several experiments the presence and response of eosinophils in LCL and DCL. Eosinophils have been neglected in the leishmaniasis contexts, probably because their significance is overshadowed by the importance of macrophages and T cells. This study address a relevant gap in the knowledge. However results here shown are rather descriptive than conclusive.

Major comments:

1) Methods section is in many points poorly described:

a) Need more specification on the eosinophil isolation: How long was centrifuged? How was done the RBC lysis? How many washes/time/g-force?

R= New information was added on lines 168-190

b) -Leishmania was cultured in the absence of antibiotics?

R=The information was added on lines 196-197.

c) -Also, in-house method for IL-8 measurement lacks important information: which capture/detection antibody was used? Please describe the protocol.

R=The information was added on lines 207-226.

d) -How was the cut-off in ELISA calculated?

R= The cut-off point information was added on lines 223-224.

e) - How was the cytometry done? What was the antibodies used and gating strategy applied?

R= Eosinophils in peripheral blood were determined by FCS and SSC by flow cytometry, this information was added on lines 133-134. The figure below represents the gating strategy. 

f) -Why is the luminescence in oxidative bursts measured as Mmvolt?

R= The Luminoskan's light detector has a photomultiplier tube (photomultiplier tube), a signal acquisition and processor system, which are in a closed compartment, where all the chemiluminescence is captured by the photomultiplier tube which sends the light signal and according to the configuration of our device (Luminoskan) the maximum light signal generated in the chemiluminescence reaction is reported in mVolts.

Line 309 – 310 “These findings demonstrate that eosinophils from patients with

DCL produce significantly more IL-6, IL-8, and IL-13 when exposed to L. mexicana

promastigotes, compared to those of LCL patients.” Cytokine measurement should have been done with unstimulated eosinophils. This is important to be sure that the cytokine production is not due to a pre-activated status. How could the authors be sure the cytokines produced were induced by parasites rather than being already activated?

R= Unstimulated eosinophils were included in all cytokine production determinations. However, they were not included in the plot because they were below the cut-off point. The information was added on lines 352-354.

Line 318-330 – To assume that eosinophils are being harmful against leishmania, the authors should show quantitatively increased death of promastigotes while in contact with Leishmania. They could use cellular death markers (Propidium iodide for example) to quantify the number of dead promastigotes in contact with eosinophil.

R= No staining with supravital dyes was performed, however, to demonstrate cell damage we are now including a new supplementary figure 2 showing Giemsa-stained viable intact parasites compared to damaged parasites incubated with eosinophils, where morphological changes and cell damage of the parasites are visible. This observation was also evidenced in the electron microscopy photographs of Figure 6, showing damage of the external membrane of the parasite. 

Lines 407-409 – It is not correct to associate IL-6 with TH2 responses. IL-6 is mainly produced by myeloid cells (mostly macrophages) and this cytokine is increased during VL together with a TH2 response, independently. IL-6 is required to a TH17 response and whether their production by eosinophils is important/significant during cutaneous leishmaniasis is yet to be shown.

R= We agree and have removed the statement regarding the association of IL-6 with a Th2 response. Instead, we added the relevance of IL-6 in the Th17 response in cutaneous leishmaniasis on lines 459-464.

Line 435: The authors says that “it is noteworthy that 70% LCL and 90% DCL were positive for IgE”, although figure 3 clearly shows (as well as is described in the text) that all patients were IgG and IgE positive. Please revise this excerpt and figure.

R= We thank the reviewer and made the correction on line 488.

Lines 442 – 444 – authors must be careful to assume “an enhanced production of cytokines” in DCL patients, since they only measured eosinophil derived cytokines and this does not mean that these cytokines are in fact increased in this patients or even related to susceptibility (for example, IFN-g is associated with resistance and it is also increased during VL). The production of these cytokines by eosinophils compared to other cells (macrophages and monocytes) might be insignificant in natural conditions.

R= We thank the reviewer and have changed the text on lines 496-501.

Figure 5 - Authors claim to demonstrate phagocitozed leishmania by eosinophil but the image has bad quality. Do authors have clear evidence of phagocytosis by eosinophils (stained slides would be enough).

R= Figure 5 was modified with a better photograph in panel 5D that shows parasites phagocyted by eosinophils. Additionally, we added a new supplementary figure 1 with more photographs showing the phagocytosis and possible intracellular degradation of the parasite. 

Authors should discuss why an innate immune cell would respond differently in DCL and LCL. Which possible mechanisms would allow eos from DCL to be more prone to respond to promastigotes than LCL?

R= We now added a possible explanation for the differential response between LCL and DCL patients on lines 452-457, including a new reference (30) where differences in genetic background between both patients have been reported.

Line 449: “their release of the cytokines IL-6, IL-8 and IL-13 promotes disease severity”: the data here shown do not support this statement.

R= We agree with the reviewer and changed the statement on lines 503-505.

Reviewer #2: The authors report on an important and up-to-now largely neglected aspect of a cellular repsonse to Leishmania infection. This study certainly will be of great interest.

Major concerns:

- The statistical method employed should be detailled in the respective Fig. legends. The authors might want to use multivariate testing.

R= The objective of the work was to compare the response between eosinophils of LCL and DCL patients and for this approach we decided use to the Mann–Whitney U-test. 

New information was added in Figure legends 1, 2 and 5, detailing the statistical significance of the figures. 

- The Fig. 5 - 7 should be analyzed using a readout that allows statistical analysis. Otherwise these findings seem very anectodal.

R= We agree and modified Figure 5, adding the panel 5E, that quantitatively describes the % of degranulation induced by parasite incubation. This result was also included in line 375-376.

- The authors should demonstrate a by FACS that their MACS-based enrichment of eosinophils worked.

R= We don´t have a FACS analysis of the enrichment of eosinophils, but include for the reviewer the figure below that shows a Giemsa stain of peripheral blood cells from patients before (A, B) and after the MACS-based enrichment (C, D). The purity of eosinophils was >99%. The information was described on lines 190-191. This purity was analyzed by a microscopical analysis of the characteristic acidophilic granules and the polymorphic nuclei as markers of eosinophils.

 Figure 1. Giemsa stain of eosinophil MACS-based enrichment. A and B show peripherical blood cells from eosinophilic patients. C and D show eosinophil enrichment. Scale bar = 20 µm.

I am sorry, but I could include the image to this text to show the enriched eosinophils.

---

## [Decision Letter · Decision Letter 1]

10 Nov 2023

PONE-D-23-05814R1Eosinophils of patients with localized and diffuse cutaneous leishmaniasis: differential response to Leishmania mexicana, with insights into mechanisms of damage inflicted upon the parasites by eosinophilsPLOS ONE

Dear Dr. Becker,

Thank you for submitting your manuscript to PLOS ONE. After careful consideration, we feel that it has merit but does not fully meet PLOS ONE’s publication criteria as it currently stands. Therefore, we invite you to submit a revised version of the manuscript that addresses the points raised during the review process.

I invite authors to revise their work but I do not anticipate another round of review if all the comments are properly addressed in the manuscript and rebuttal letter.

We look forward to receiving your revised manuscript.

Kind regards,

Vyacheslav Yurchenko, Ph.D.

Academic Editor

PLOS ONE

Journal Requirements:

Reviewers' comments:

Reviewer's Responses to Questions

**Comments to the Author**

1. If the authors have adequately addressed your comments raised in a previous round of review and you feel that this manuscript is now acceptable for publication, you may indicate that here to bypass the “Comments to the Author” section, enter your conflict of interest statement in the “Confidential to Editor” section, and submit your "Accept" recommendation.

Reviewer #1: All comments have been addressed

Reviewer #3: All comments have been addressed

2. Is the manuscript technically sound, and do the data support the conclusions?

Reviewer #1: Yes

Reviewer #3: Yes

3. Has the statistical analysis been performed appropriately and rigorously? 

Reviewer #1: Yes

Reviewer #3: Yes

4. Have the authors made all data underlying the findings in their manuscript fully available?

Reviewer #1: Yes

Reviewer #3: Yes

5. Is the manuscript presented in an intelligible fashion and written in standard English?

Reviewer #1: Yes

Reviewer #3: Yes

6. Review Comments to the Author

Reviewer #1: The entire manuscript has been improved, and the questions were mostly solved. However a final adjustment must be done. The cytometry plot is very strange and unusual. Is there a better figure that could be alternatively used? I recommend that the authors report more details about the method used: how flow cytometry as performed? How RBC were lysed? How do they processed the blood? Important to mention which equipment was used for flow cytometry.

Reviewer #3: The study by Salaiza-Suazo et al. addresses the important yet mostly unknown aspect of eosinophils in the immune response against CL in the new world. Furthermore, this study uses patient samples to compare various parameters of eosinophil responses among those experiencing the localized (LCL) versus the severe diffuse (DCL) form of disease. Overall, the study is a useful step towards elucidating the role of eosinophils in L. mexicana infection as well as in sorting out differences contributing to LCL versus DCL.

I consider that in the R1 version of the manuscript, the authors have appropriately addressed the initial set of concerns and suggestions raised by the reviewers. There are no major concerns.

Minor concerns: A total of 25 minor concerns have been added as comments in the pdf version of PONE-D-23-05814R1.

7. PLOS authors have the option to publish the peer review history of their article (what does this mean?). If published, this will include your full peer review and any attached files.

Reviewer #1: No

Reviewer #3: No

---

## [Author Response · Author response to Decision Letter 1]

16 Dec 2023

Dear Editor, 

we thank the reviewers for the valuable comments that have improved the manuscript. We have made the suggested corrections and addressed the points raised by the reviewer and hope the manuscript is now suitable for publication.

Corrections are listed according to the lines in the original manuscript, which are shown in the parentheses.

1.- (Line 34) extra spaces were removed in line 34.

2.- (Line 98) “mexicana” was corrected to lower case in line 97. 

3.- (Line 188-190) Font size was corrected lines 189-190.

4.- (Line 244) the text was corrected to “intracellular parasites” in line 375. 

5.- (Line 289) The age range of LCL patients was included in line 289.

6.- (Line 296) the correction was made changing 1B to “1A” in line 295.

7.- (Lines 300, 304) New information and 4 new references (49-52) suggested by the reviewer were included. Furthermore, an additional reference (53) on male susceptibility was also included. The new text included in the manuscript is found in lines 508-522.

8.- (Line 306) We have modified figure 2, including a new graph of eosinophil counts in tissues (2B) of males and females with LCL and DCL. Due to the modifications of Figure 2, changes were made in line 308, where the word “Fig. 2A” was inserted and in line 314. Furthermore, the Figure legend for Figure 2 was modified in lines 816-820. Unfortunately, we do not have all the ages of the patients available at the moment.

9.- (Lines 318-320) A text describing results in Figure 2B was included in lines 815-819.

10.- (Line 353) the correction was made in line 354.

11.- (Line 374) The word “amastigote” was substituted for “parasites” in line 375.

12.- (Line 376) The correction was made and now reads Fig 5“E” in line 377.

13.- (Line 402) The correction was made stating “1-5 %” in line 403, ref. #4 is cited.

14.- (Lines 404 and 405) the corrections were made in lines 405 and 406.

15.- (Line 406) The comment was addressed in lines 508-522.

16.- (Line 451) 

#1. The observation was addressed in lines 508-522 of the discussion and 4 new references (49-52) suggested by the reviewer were included. Furthermore, an additional reference (53) on male susceptibility was also included. The new text added to the manuscript is found in lines 508-522.

#2. The observation was addressed in the discussion lines 479-483. The corrected Figure 2B now includes information on eosinophilia and male patients with localized or diffuse cutaneous leishmaniasis and 2 new references (43, 44) were added.

17.- (Line 479) The “L” was corrected and now reads IL-13 in line 485

18.- (Line 503) The sentence was corrected in lines 523-527.

19.- (Line 758) Figure Legend 1 was corrected (lines 806-809). With regard to the suggestion of illustrating males vs females in panel 1A, we believe that the information is already presented in panel 1B, and prefer not to overload panel A.

20) (line 760) The word “dot” was substituted by “symbol” in line 810.

21)- (line 769) We have included more information in new Figure 2B.

22) (Line 800) The additional space was eliminated in line 859.

23) (Line 807) The information of the scale bar was included in line 861.

24) (Line 824) The information for supplementary Figure 1 was added in line 878.

25) (Line 830) The information for supplementary Figure 2 was added in line 885-886.

26) The symbols were modified in Fig.1A and the Figure legend was adjusted (lines 806-810). 

Thank you very much.

---

## [Editor Report · Decision Letter 2]

20 Dec 2023

Eosinophils of patients with localized and diffuse cutaneous leishmaniasis: differential response to Leishmania mexicana, with insights into mechanisms of damage inflicted upon the parasites by eosinophils

PONE-D-23-05814R2

Dear Dr. Becker,

We’re pleased to inform you that your manuscript has been judged scientifically suitable for publication and will be formally accepted for publication once it meets all outstanding technical requirements.

Kind regards,

Vyacheslav Yurchenko, Ph.D.

Academic Editor

PLOS ONE
---

## [Editor Report · Acceptance letter]

7 Feb 2024

PONE-D-23-05814R2 

PLOS ONE

Dear Dr. Becker, 

I'm pleased to inform you that your manuscript has been deemed suitable for publication in PLOS ONE. Congratulations! Your manuscript is now being handed over to our production team.

Kind regards, 

on behalf of

Prof. Vyacheslav Yurchenko 

Academic Editor

PLOS ONE